# Application of Comparative Genomics for the Development of PCR Primers for the Detection of Harmful or Beneficial Microorganisms in Food: Mini-Review

**DOI:** 10.3390/foods14061060

**Published:** 2025-03-20

**Authors:** Sang-Soon Kim

**Affiliations:** School of Animal & Food Sciences and Marketing, Konkuk University, Seoul 05029, Republic of Korea; quks5120@konkuk.ac.kr; Tel.: +82-010-9184-5269

**Keywords:** comparative genomics, pan-genome analysis, PCR primers, foodborne pathogen, detection

## Abstract

Gene markers are widely utilized for detecting harmful and beneficial microorganisms in food products. Primer sequences targeting the 16S rRNA region, recognized as a conserved region, have been conventionally employed in PCR analyses. However, several studies have highlighted limitations and false-positive results associated with the use of these primer sequences. Consequently, pan-genome analysis, a comparative genomic approach, has been increasingly applied to design more selective gene markers. This mini-review explores the application of pan-genome analysis in developing PCR primers for the detection of harmful microorganisms, such as *Salmonella*, *Cronobacter*, *Staphylococcus*, and *Listeria*, as well as beneficial microorganisms like *Lactobacillus*. Additionally, the review discusses the applicability, advantages, limitations, and future directions of pan-genome analysis for primer design. A comparative overview of bioinformatics tools, recent trends, and verification methods is also provided, offering valuable insights for researchers interested in leveraging pan-genome analysis for advanced primer design.

## 1. Introduction

Comparative genomics, which examines the similarities and differences among the genomes of various organisms, has become widely utilized across diverse scientific disciplines, propelled by advancements in next-generation sequencing technologies. Comparing genomes provides insights into evolutionary relationships, functional gene identification, genetic variation discovery, and progress in biomedical research. Among the methods used in comparative genomics, pan-genome analysis is particularly prominent [1]. Pan-genome analysis categorizes genomic content into two components: the core genome, shared by all strains and crucial for growth and survival, and the accessory genome, unique to specific strains, which sheds light on genomic adaptability, specialized lifestyles, and evolutionary dynamics [2]. This approach is particularly effective for understanding the genetic diversity and evolutionary clades of pathogenic bacteria. Recently, pan-genome analysis has found applications in various domains, including medicine, public health, and the food industry [3]. For example, it has been used to discover novel antiphage defense systems [4], enable reverse vaccinology approaches [5,6], and identify potential vaccine targets [7]. Additionally, pan-genome analysis has proven valuable for tracking evolutionary trajectories and lineage relationships during bacterial outbreak investigations [8], as well as elucidating new evolutionary dynamics within bacterial serotype groups [9]. Various software tools are available for pan-genome analysis, including Pan-Genome Analysis Pipeline-Extended (PGAP-X), Roary, the Bacterial Pan Genome Analysis (BPGA) pipeline, EDGAR, seq-seq-pan, and panX, each offering unique features (Table 1). PGAP-X allows for the visualization of whole-genome alignments, genetic variation analysis, functional annotation, and the identification of core and accessory genes [10]; however, it requires advanced bioinformatics expertise for effective use. Roary is a fast and efficient tool for pan-genome visualization, specifically for prokaryotes, but has lower sensitivity when analyzing highly divergent genomes. The BPGA pipeline facilitates phylogenetic generation predictions and the identification of unique gene presence or absence [11], but has limited visualization capabilities. EDGAR, a web-based tool, focuses on providing intuitive visualizations for comparative genomics with limited computational power and customization efficacy. Seq-seq-pan employs a graph-based visualization approach, which demands expertise in graph processing. Lastly, panX integrates phylogenetic and genomic analyses with interactive visualization, offering an intuitive interface for exploring pan-genomic data.

The rapid and accurate detection of pathogenic bacteria in food samples is a critical area of research. Various detection methods have been developed, including conventional culture-based techniques and alternative approaches such as polymerase chain reaction (PCR), isothermal amplification, enzyme-linked immunosorbent assay (ELISA), bacteriophage amplification, and gold nanoparticle aggregation methods [16]. Among these alternatives, real-time PCR has become a widely used tool for detecting foodborne pathogens due to its sensitivity and specificity. Traditionally, the 16S rRNA gene has been used as a marker for PCR analysis, as it is considered a highly conserved region in bacterial genomes. However, numerous studies have reported false-negative and false-positive results when using primers designed for the 16S rRNA region [17,18]. These findings underscore the need for alternative markers to improve PCR-based detection reliability. Comparative genomics provides a promising approach for designing new, more specific primers. By analyzing genetic variability and identifying unique gene regions, this method offers the potential to enhance the accuracy and selectivity of PCR-based pathogen detection.

In this mini-review, recent studies on the application of pan-genome analysis for detecting foodborne pathogens, such as *Salmonella*, *Cronobacter*, *Staphylococcus*, and *Listeria*, as well as beneficial bacteria like *Lactobacillus*, are introduced and discussed. The target species, pan-genome analysis tools, detection methods, and key findings are summarized and compared in the tables.

## 2. Development of PCR Primer Based on Comparative Genomics for the Detection of *Salmonella*

*Salmonella* is a well-known foodborne pathogen responsible for numerous outbreaks in South Korea and the United States [19]. The species of *Salmonella* that cause foodborne illness are primarily categorized into two groups: *Salmonella enterica* and *Salmonella bongori*. Among these, *S. enterica* is the primary species associated with foodborne illness, further divided into six subspecies and more than 2600 serotypes. These serotypes are classified into groups A, B, C1, C2, D, and E based on the O antigen. Notably, *Salmonella enterica serova Typhimurium*, *S. Enteritidis*, *S. Newport*, and *S. Montevideo* are frequently implicated in foodborne outbreaks. As a result, researchers are focusing on the development of PCR primers specifically targeting these notorious serotypes (Table 2).

For example, Ref. [20] identified a gene target for *Salmonella enterica* serovar Montevideo through pan-genome analysis. In their study, primer-probe sets for detecting both *Salmonella enterica* and *S. enterica* serovar *Montevideo* were developed using the panX tool, based on data from 706 *S. enterica* strains, including 23 strains of *S. Montevideo*. Furthermore, the applicability of these primers was assessed in food samples, including tomato, raw chicken meat, red pepper, and black pepper. The results indicated that the developed realtime PCR is more effective to detect foodborne pathogens in food samples compared to the conventionally used XLD media. Considering the fact that the detection and inactivation of *Salmonella* spp. in red and black pepper has been particularly challenging, the developed primers showed potential for effective use in these samples. Ref. [21] designed specific primers for rapid detection of the E serogroup (Weltevreden, London, Meleagridis, and Senftenberg) using Roary and validated the primers in artificially contaminated food samples, including chicken, pork, beef, eggs, fish, and vegetables. The study verified the sensitivity and selectivity of the primers through conventional PCR, with further research needed to evaluate their application in real-time PCR. In a related study, Ref. [22] identified the *ssaQ* gene as a target for *Salmonella* detection using Roary and demonstrated that loop-mediated isothermal amplification (LAMP) exhibited higher sensitivity than conventional PCR with the selected primers. A comparison with real-time PCR could provide further insights for researchers.

The application of BPGA tool for primer design has been reported in previous studies. For instance, Ref. [23] developed a gene marker specific for *Salmonella* Infantis (*SIN_02055*) by profiling 60 *Salmonella* serovars using the BPGA tool. The authors demonstrated that the designed marker accurately distinguished *S.* Infantis with 100% specificity. In another study by the same group [24], novel gene markers for detecting 60 *Salmonella* serovars were designed with BPGA and validated through real-time PCR. These studies highlight the flexibility of pan-genome analysis in customizing target ranges, allowing for the targeting of multiple serovars or a single serovar, depending on the genomic comparison and primer selection. This approach is particularly valuable for addressing foodborne outbreaks caused by specific *Salmonella* serovars in particular food items.

## 3. Development of PCR Primer Based on the Comparative Genomics for the Detection of *Cronobacter*

*Cronobacter* species are Gram-negative, facultative anaerobic bacteria within the family Enterobacteriaceae, known to be opportunistic pathogens, particularly affecting immunocompromised individuals. Among the species, *C. sakazakii* has been implicated in serious outbreaks, particularly in infants, and was classified as an *Enterobacter* species until 2008 [25]. Traditionally, primers targeting the *ompA*, *gluA*, and *wzx* genes have been used for the detection of *C. sakazakii* via real-time PCR. However, Ref. [26] reported that the target based on 16S rRNA region could not distinguish closely related *C. sakazakii* strains. This finding indicates the need for a new approach in real-time PCR analysis to develop more sensitive and specific primers. In this context, comparative genomics-based primer design offers an alternative for more accurate bacterial detection. Recent studies have reported the detection potential of various bacteria through pan-genome analysis, as summarized in Table 3.

For instance, Ref. [27] proposed a new primer-probe set for detecting *C. sakazakii* based on comparative genomics. In their study, a gene annotated as type 1 fimbrial protein, among 16 candidate genes, was selected as the target of primer, and the detection efficacy was validated in food samples such as powdered infant formula, powdered infant formula containing *Lactobacillus*, and milk. Similarly, Ref. [28] targeted the *fimG* (type 1 fimbrial protein) and *lpfA_1* (fimbrial protein) genes for *C. sakazakii* detection, demonstrating that multiplex PCR was effective for detecting multiple *Cronobacter* species. Ref. [29] conducted large-scale comparative genomic analysis to identify novel genus- and species-specific genes for the detection of *C. sakazakii*, *C. malonaticus*, and *C. turicensis*, but pan-genome analysis was not employed in this study. While this approach is interesting, it may lead to false-positive results due to low selectivity in food samples. Collectively, there is a relatively limited number of studies on the use of pan-genome analysis for detecting *Cronobacter* species in food. Therefore, further research is necessary to develop additional primer-probe sets for detecting clinical *Cronobacter* in food samples using pan-genome analysis. These approaches are crucial for ensuring the safety of powdered infant formula.

## 4. Development of PCR Primer Based on the Comparative Genomics for the Detection of *Staphylococcus* spp.

*Staphylococcus* spp. are among the most frequently detected foodborne pathogens in both food and environmental samples, often forming biofilms on various food contact surfaces [30]. Various efforts have been made to identify selective primer markers for the detection of *Staphylococcus* spp (Table 4). For example, Ref. [31] proposed four novel target genes: *comFA* for *S. aureus*, *group_14348* for *S. epidermidis*, *group_26190* for *S. haemolyticus*, and *group_26478* for *S. hominis*, and demonstrated their sensitivity, specificity, and efficiency using 100 samples. Similarly, Ref. [17] identified four novel molecular targets through pan-genome analysis: GntR family transcriptional regulator for *S. aureus*, *phosphomannomutase* for *S. epidermidis*, FAD-dependent urate hydroxylase for *S. capitis*, and Gram-positive signal peptide protein for *S. caprae*. In this study, verification was conducted using various types of food samples, including beef, pork, lettuce, cucumber, raw milk, and fermented fish. These food items are particularly important for detection studies, as certain components in food can interfere with the efficacy of PCR analysis.

Targets for the detection of other *Staphylococcus* species have also been investigated. For example, Ref. [32] proposed a specific primer target for detecting *S. pseudintermedius*, an opportunistic pathogen in dogs, cats, and humans. The suggested real-time PCR analysis demonstrated that the designed primer effectively detects *S. pseudintermedius* with high specificity. Recently, an interesting study by Ref. [33] explored novel target genes for the detection of *S. argenteus* in food through pan-genome analysis. The authors reported that *S. argenteus*, a newly identified species distinct from *S. aureus*, poses a potential threat to human health. Their research group conducted pan-genome analysis of 693 *Staphylococcus* strains, including 227 *S. aureus* and 118 *S. argenteus* strains sequenced in their laboratory. This approach highlights the potential of pan-genome analysis to suggest primer-probe sets for newly isolated or reclassified species, demonstrating its effective application in food safety.

**Table 4 foods-14-01060-t004:** Designation of PCR primers for the detection of *Staphylococcus* using comparative genomics.

Species	Pan-Genome Analysis Tools (Version)	Detection Method	Main Results	Year	Reference
*S. aureus* *S. capitis* *S. caprae* *S. epidermidis*	BPGA pipeline (v1.3)	Real-time qPCR	- Four new molecular targets were mined based on pan-genome analysis - The developed detection method successfully identified strains isolated from various food matrixes (chicken, beef, pork, fish, salted fish, and raw milk).	2021	[17]
*S. aureus*, *S. epidermidis*, *S. haemolyticus* *S. hominis*	Roary	Real-time qPCR	- Gene targets were selected based on pan-genome analysis, and the gene-based detection method enabled rapid, sensitive, and accurate detection of *Staphylococcus* spp.	2022	[31]
*S. argenteus*	Roary	Conventional PCR Realtime qPCR	- Pan genome analysis was performed for 693 *Staphylococci* strains - 20 specific genes were found and five genes were validated	2024	[33]
*S. pseudintermedius*	Roary (v3.5.6)	Realtime qPCR	- Specific target for the detection of *S. pseudintermedius* was suggested - Specificity of the suggested primer was verified with Realtime qPCR	2017	[32]

## 5. Development of PCR Primer Based on the Comparative Genomics for the Detection of *Listeria*

*Listeria* is a Gram-positive bacterium, and *L. monocytogenes* is known as one of the most notorious foodborne pathogens, particularly dangerous to fetuses. Various studies have been published on methods to inactivate the pathogen [34,35]. *L. monocytogenes* can grow at low temperatures and can be transmitted from mother to fetus through vertical transmission [36]. Therefore, detecting *L. monocytogenes* in food samples is crucial, and various methods have been reported (Table 5). In food safety, distinguishing between *L. monocytogenes* and *L. innocua* has been challenging due to the genetic and phenotypic similarities between the two species. Recently, Ref. [18] developed a duplex real-time PCR method for the detection of *L. innocua* and *L. monocytogenes* based on comparative genomics. In this study, they demonstrated that the newly developed primer could replace the traditionally used *iap* gene, which had shown false-positive results. In another study, Ref. [37] designed a multiplex PCR assay for the simultaneous detection of three *L. monocytogenes* lineages (I, II, and III) and five major serotypes (1/2a, 1/2b, 1/2c, 4b, and 4c).

In the food industry, detecting *L. monocytogenes* in mushroom samples has gained attention due to multistate outbreak cases in the United States linked to the consumption of enoki mushrooms imported from South Korea [38]. As a result, several studies have focused on developing PCR primers for the detection of *Listeria* in mushroom samples. For example, Ref. [39] developed a multiplex PCR for the identification of pathogenic *Listeria* species (*L. monocytogenes* and *L. ivanovii*) in fresh *Flammulina velutipes* mushrooms. The study showed that the multiplex PCR method was highly effective and consistent with traditional culture-based techniques. Similarly, Ref. [40] designed specific gene markers for detecting *L. monocytogenes* and *L. monocytogenes* CC8, validating the method with 12 mushroom samples using both multiplex PCR and high-resolution melting qPCR. These approaches contribute to reducing the risk of *L. monocytogenes* contamination in mushroom samples.

**Table 5 foods-14-01060-t005:** Designation of PCR primers for the detection of *Listeria* spp. using comparative genomics.

Species	Pan-Genome Analysis Tools (Version)	Detection Method	Main Results	Year	Reference
*L. monocytogenes* *L. ivanovii* Nonpathogenic *Listeria*	Roary (v3.11.2)	Conventional PCR Multiplex PCR	- Target of *L. monocytogenes, L. ivanovii*, and non-pathogenic *Listeria* was suggested - Suggested target was verified with conventional and multiplex PCR	2021	[39]
*Listeria monocytogenes* lineage (I, II, III) *Listeria monocytogenes* serotypes (1/2a, 1/2b, 1/2c, 4b, 4c)	Roary (v3.11.2)	Conventional PCR Multiplex PCR	- New target genes for the detection of *L. monocytogenes* were investigated using pan-genome analysis - Multiplex PCR analysis with designed primer distinguish three lineages (I, II, and III) and five major serotypes (1/2a, 1/2b, 1/2c, 4b, and 4c) of *L. monocytogenes* simultaneously.	2021	[37]
*Listeria monocytogenes* *Listeria innocua*	Roary (v3.13.0)	Selective media (OAB) Multiplex PCR	- Developed primer-probe based on pan genome analysis showed 100% specificity and selectivity for the detection of *L. monocytogenes* - Color pigments of food sample affect the results of real-time PCR	2024	[18]
*L. monocytogenes* *L. monocytogenes* clonal complex 8 (CC8)	Roary (v3.11.2)	Multiplex PCR High-resolution meting qPCR (HRM)	- Primers for detection of *L. monocytogenes* and CC8 strain were suggested - The detection limits were 2.1 × 10^3^ and 2.1 × 10^0^ CFU/mL for multiplex PCR and HRM qPCR, respectively. - Feasibility of the suggested primers were evaluated with 12 mushroom samples	2024	[40]

## 6. Development of PCR Primer Based on the Comparative Genomics for the Detection of *Lactobacillus*

*Lactobacillus* is a well-known beneficial microorganism widely used in various food products with numerous previous studies [41,42,43]. The type and number of probiotics are often labeled on food products, making the detection of probiotics like *Lactobacillus* important in the food industry. Several studies have focused on identifying gene markers for probiotic detection (Table 6). For example, Ref. [44] reported that *Latilactobacillus sakei* consists of four closely related species and proposed novel markers for distinguishing the *L. sakei* group and its subspecies. These PCR primers were validated with 106 strains isolated from fermented foods. Similarly, genetic markers for *Lacticaseibacillus zeae* [45] and *Lactobacillus delbrueckii* [46] were developed. An interesting study by the same group [47] who used selective markers based on comparative genomics to examine probiotics in food products. The authors suggested that the developed method could be effectively used to verify the labeling of probiotic products. Given the challenges in rapidly identifying labeling fraud in the food safety sector, this approach is expected to be highly effective in the near future.

## 7. Conclusions and Future Remarks

In this mini-review, the application of comparative genomics for the development of PCR primers targeting foodborne pathogens and other microorganisms relevant to the food industry was introduced and discussed. Among the bioinformatics tools, BPGA, Roary, and panX have been frequently employed in pan-genome analysis, as highlighted in the reviewed studies. For *Salmonella* detection, pan-genome analysis has been especially useful for designing serotype-specific primers, as certain serotypes, such as *S.* Montevideo and *S.* Senftenberg, are significant in the context of food safety. Research on *Cronobacter* primer development has been relatively limited compared to other pathogens, and further studies are needed to address the impact of *C. sakazakii*, especially in food samples like powdered infant formula. In the case of *Staphylococcus*, novel primer sequences have been designed for detecting less-studied species such as *S. argenteus* and *S. pseudintermedius*, which have been relatively neglected compared to *S. aureus*. For *Listeria*, primer development has mainly focused on *L. monocytogenes*, particularly for use in mushroom samples due to the multistate outbreaks associated with enoki mushrooms imported from South Korea. Finally, pan-genome analysis has also been applied to the development of primer-probes for detecting probiotics in food samples, especially *Lactobacillus* species. As demonstrated in this review, the development of new and more efficient PCR primers holds great promise in the field of food safety, and comparative genomics is a valuable tool in advancing PCR primer development.

On the other hand, a key limitation of applying comparative genomics is its reliance on high-quality genomic data and substantial computational resources. The accuracy of pan-genome analysis can vary depending on the available genomic datasets, requiring periodic updates to maintain reliability. Additionally, the development of specific gene targets and the validation of their sensitivity and selectivity demand expertise in advanced bioinformatics. In food samples, the presence of inhibitors may affect the applicability of designed primers, necessitating further optimization. Therefore, the role of food safety engineers with expertise in both microbiological experimentation and bioinformatics will become increasingly vital in the near future.

## Figures and Tables

**Table 1 foods-14-01060-t001:** Properties, advantages, and limitations of software tool for pan-genome analysis (PGAP-X, Roary, BPGA, EDGR, Seq-Seq-Pan, and panX).

Tool	Property	Advantage	Limitation	Reference
PGAP-X	Scalable and modular architecture	- High scalability - Suitable for large dataset and customization	- High computational demand - High bioinformatics skill demand	[10]
Roary	Core genome analysis with pre-clustering approach (High speed)	- Fast and efficient - Visualization of output data	- Limited to bacterial genome - Low sensitivity in highly divergent genome	[12]
Bacterial pan genome anlysis pipeline (BPGA)	Incorporation of functional annotation and orthologous group clustering	- Identification of functional insight - Ease to use	- Limited scalability - Demand of high-quality genome assemblies	[11]
EDGAR	Web-based tool focusing on visualization	- Intuitive (web interface) - Comprehensive visualization - Small genome set handling	- Limited scalability - Dependency on web interface	[13]
Seq-Seq-Pan	Visualization of genome variation with graph-based method	- Graph-based visualization - Identification of genomic relationships	- High computational demand - Requires high skill on the graph-processing	[14]
panX	Integration of phylogenetic and genomic visualization	- Interactive visualization - Combination of evolutionary context with genomic insight	- Limited scalability	[15]

**Table 2 foods-14-01060-t002:** Designation of PCR primers for the detection of *Salmonella* spp. using comparative genomics.

Species	Pan-Genome Analysis Tools (Version)	Detection Method	Main Results	Year	Reference
*S.* Montevideo	panX	Real-time qPCR	- The primer-probe sets were developed to detect *S.* Montevideo using panX program - The developed primer-probe showed high sensitivity and selectivity - Food application was conducted with raw chicken meat, red pepper, and black pepper	2022	[20]
E serogroup (*S.* Weltevreden, *S.* London, *S.* Meleagritis, *S.* Senftenberg)	Roary (v3.11.2)	Conventional PCR	- New target for the detection of *Salmonella* E serogroup was suggested - Food application was conducted for artificially contaminated food (chicken, pork, beef, eggs, fish, vegetables)	2021	[21]
*Salmonella* genus (Include *S. bongori*) *Salmonella enterica*	Roary	Conventional PCR Loop-mediated isothermal amplification (LAMP)	- *ssaQ* gene was selected as the target of *Salmonella* - Sensitivity of LAMP method was higher than conventional PCR-based method with selected primer	2021	[22]
*Salmonella* Infantis	Bacterial Pan-Genome Analysis Pipeline, (BPGA, v1.3)	Real-time qPCR	- Gene marker specific for *Salmonella* Infantis (SIN_02055) was selected by profiling 60 *Salmonella* serovars - Designed marker distinguishes *S.* Infantis with 100% accuracy	2020	[23]
*Salmonella* 60 serovars	BPGA	Real-time qPCR	- The novel gene markers for 60 serovars of *Salmonella* were explored using pangenome analysis - PCR analysis verified that the designed gene marker distinguishes the 60 most common *Salmonella* serovars.	2021	[24]

**Table 3 foods-14-01060-t003:** Designation of PCR primers for the detection of *Cronobacter* using comparative genomics.

Species	Pan-Genome Analysis Tools (Version)	Detection Method	Main Results	Year	Reference
*Cronobacter sakazakii*	panX	Real-time PCR (qPCR)	- A new primer-probe set for detecting *C. sakazakii* was designed - Efficacy of realtime PCR was verified by comparing with the selective medium.	2022	[27]
*C. sakazakii*, *C. malonaticus*, *C. dublinensis*, *C. turicensis*	Roary (v3.11.2)	Conventional PCR, Multiplex PCR	*-* Primer-probe targeting *Cronobacter* species (*sakazakii*, *malonaticus*, *deblinensis*, *turicensis*) was designed. - PCR assays were identified to be specific and sensitive in the detection of *Cronobacter*.	2021	[28]

**Table 6 foods-14-01060-t006:** Designation of PCR primers for the detection of probiotics using comparative genomics.

Species	Pan-Genome Analysis Tools	Detection Method	Main Results	Year	Reference
*Latilactobacillus sakei* group (*L. sakei*, *L. curvatus*, *L. graminis*, and *L. fuchuensis*)	BPGA pipeline v1.3	Real-time PCR	- A new marker for PCR detection and identification of *L. sakei* group species and *L. sakei* subspecies was identified through comparative pan-genomic analysis. - The primer pairs were designed for each marker and qualitative and quantitative identification demonstrated that the marker gene can be used as alternative to 16S rRNA gene.	2021	[44]
*Lacticaseibacillus zeae*	BPGA	Real-time PCR	- A unique gene of *L. zeae* was identified through pan-genome analysis	2021	[45]
*Lactobacillus delbrueckii* (*L. delbrueckii* subsp. *bulgaricus*, *L. delbrueckii* subsp. *lactis*, *L. delbrueckii* subsp. *delbrueckii*)	BPGA pipeline v1.3	Real-time PCR	- A specific primer pair for accurate identification and identification of *L. delbrueckii* subspecies was designed based on pan-genome analysis. - The results showed 100% specificity for each subspecies and were able to distinguish 44 different lactic acid bacteria from each subspecies.	2021	[46]

## Data Availability

No new data were created or analyzed in this study. Data sharing is not applicable to this article.

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
