# Peer review of "Application of Comparative Genomics for the Development of PCR Primers for the Detection of Harmful or Beneficial Microorganisms in Food: Mini-Review"

_foods, 2025, doi:10.3390/foods14061060_

Round 1

Reviewer 1 Report

Comments and Suggestions for Authors

This mini-review exhibits some innovation and significance. In this review, the recent studies on the application of pan-genome analysis for detecting foodborne pathogens (including Salmonella, Cronobacter, Staphylococcus, and Listeria) as well as beneficial microorganisms like Lactobacillus were introduced and discussed. This research direction holds significant application prospects in the field of food safety, capable of markedly enhancing the accuracy and efficiency of food detection. In addition, this review provides an overview of the application of various bioinformatics tools (such as PGAP-X, Roary, BPGA, etc.) and conducts a comparative analysis of their respective advantages and disadvantages, which offered valuable insights for researchers interested in leveraging pan-genome analysis for advanced primer design. The manuscript was well written need the following modifications:

Major:

  1. The manuscript dedicates substantial space to elaborating the advantages and potential of pan-genome-based specific primer PCR methodology for detecting harmful microorganisms. However, it inadequately addresses objective evaluations of this method's limitations, including instrumentation costs, reagent/supply costs, time consumption, site-specific infrastructure requirements, and portability limitations. It is recommended to present a more balanced and comprehensive discussion of this new approach.
  2. The manuscript lacks a systematic comparison between the specific primer PCR detection method (the focal technique) and conventional detection approaches using identical samples. Critical parameters such as sensitivity, detection limits, accuracy, error margins, and cost-effectiveness require explicit benchmarking. Providing a comprehensive comparative table would significantly enhance the scholarly value of this review.
  3. The "Conclusions and Future Remarks" section is underdeveloped, with future perspectives confined to a single concluding sentence. It is strongly advised to restructure this section into two distinct paragraphs, dedicating comparable length to detailed, constructive forward-looking discussions. This expansion would substantially strengthen the manuscript's scientific impact.

Minor:

  1. Table 1 currently lacks supporting citations for its presented data. Appropriate references should be provided to validate the information sources.
  2. All the tables’ presentation exhibit excessive row spacing, resulting in unnecessary page consumption. Formatting adjustments to achieve compact yet legible table layouts would improve visual coherence and professional presentation.
  3. Did the authors want to provide Figure 1, Table 4, Table 5 and Table 6 as supplementary materials? If they are not supplementary materials, they should be properly cited in the main text and strategically positioned within corresponding content sections.

Author Response

Comments 1: This mini-review exhibits some innovation and significance. In this review, the recent studies on the application of pan-genome analysis for detecting foodborne pathogens (including SalmonellaCronobacterStaphylococcus, and Listeria) as well as beneficial microorganisms like Lactobacillus were introduced and discussed. This research direction holds significant application prospects in the field of food safety, capable of markedly enhancing the accuracy and efficiency of food detection. In addition, this review provides an overview of the application of various bioinformatics tools (such as PGAP-X, Roary, BPGA, etc.) and conducts a comparative analysis of their respective advantages and disadvantages, which offered valuable insights for researchers interested in leveraging pan-genome analysis for advanced primer design. The manuscript was well written need the following modifications:

  Response 1: Thank you for your effort to review the manuscript. Please review the revised manuscript once more.

Comments 2: The manuscript dedicates substantial space to elaborating the advantages and potential of pan-genome-based specific primer PCR methodology for detecting harmful microorganisms. However, it inadequately addresses objective evaluations of this method's limitations, including instrumentation costs, reagent/supply costs, time consumption, site-specific infrastructure requirements, and portability limitations. It is recommended to present a more balanced and comprehensive discussion of this new approach.

Response 2: Thank you for your insightful comment. The limitation of the pan genome approach was not indicated comprehensively as you mentioned. Therefore, the limitations were added in the manuscript as you recommended (L 257-265).

Comments 3: The manuscript lacks a systematic comparison between the specific primer PCR detection method (the focal technique) and conventional detection approaches using identical samples. Critical parameters such as sensitivity, detection limits, accuracy, error margins, and cost-effectiveness require explicit benchmarking. Providing a comprehensive comparative table would significantly enhance the scholarly value of this review.

Response 3: Thank you for your insightful comment. The systematic comparison would be invaluable as you mentioned, but the difference would be various depending on the type of pathogen and other conditions. Moreover, the comparisons between the conventional method and the developed method were not indicated in many of cited papers. Therefore, we described the improvement by using the specific primer compared to the conventional detection approaches in the text (L 105-107).

Comments 4: The "Conclusions and Future Remarks" section is underdeveloped, with future perspectives confined to a single concluding sentence. It is strongly advised to restructure this section into two distinct paragraphs, dedicating comparable length to detailed, constructive forward-looking discussions. This expansion would substantially strengthen the manuscript's scientific impact.

Response 4: Thank you for your insightful comment. The “conclusions and future remarks” section was divided into two distinct paragraphs as you recommended. Please check the revised manuscript (L 237-265).

Comments 5: Table 1 currently lacks supporting citations for its presented data. Appropriate references should be provided to validate the information sources.

Response 5: Thank you for your valuable comment. The sources of Table 1 were added as you recommended (Table 1 and L 297-306).

Comments 6: All the tables’ presentation exhibit excessive row spacing, resulting in unnecessary page consumption. Formatting adjustments to achieve compact yet legible table layouts would improve visual coherence and professional presentation.

Response 6: Thank you for your valuable comment. The row spaces in the tables were shortened to show more compact table presentation (Table 1-6). 

Comments 7: Did the authors want to provide Figure 1, Table 4, Table 5 and Table 6 as supplementary materials? If they are not supplementary materials, they should be properly cited in the main text and strategically positioned within corresponding content sections.

Response 7: Thank you for the valuable comment. The Figure 1, Table 4-6 were moved to the appropriate place as you recommended.

Reviewer 2 Report

Comments and Suggestions for Authors

In this mini-review, the author describe as genomic approach is applied to design more selective gene markers to detect diverse harmful microorganisms, in specific this manuscript describe some references related to Salmonella, Cronobacter, Staphylococcus, and Listeria, as well as beneficial microorganisms like Lactobacillus. The minireview is very interesting, the information is well-performed and well-written.

I suggest the following revisions to further improve this minireview.

Here is my observation:

1.- I suggest change the title, because the methodology for the application of comparative genomic is not described in the text, it only is mentioned in some tables and text.

2.- Please revise the scientific names for some bacteria, specifically for specie names, for example: S. Typhimurium, S. Enteritidis, S. Newport, and S. Montevideo.

3.- Novel gene markers are important to describe for these bacterial species described. Because for Listeria and Lactobacillus for example only a short information is mentioned.

Author Response

Comments 1: In this mini-review, the author describe as genomic approach is applied to design more selective gene markers to detect diverse harmful microorganisms, in specific this manuscript describe some references related to SalmonellaCronobacterStaphylococcus, and Listeria, as well as beneficial microorganisms like Lactobacillus. The minireview is very interesting, the information is well-performed and well-written. I suggest the following revisions to further improve this minireview. Here is my observation:

Response 1: Thank you for your effort to review the manuscript. Please check the revised manuscript once more.

Comments 2: 1.- I suggest change the title, because the methodology for the application of comparative genomic is not described in the text, it only is mentioned in some tables and text.

Response 2: Thank you for your kind suggestion, but every text of this manuscript is about the application of comparative genomics. Therefore, all sub-titles are revised to represent the application of comparative genomics (L 87; 129; 159; 188; 218).

Comments 3: 2.- Please revise the scientific names for some bacteria, specifically for specie names, for example: S. Typhimurium, S. Enteritidis, S. Newport, and S. Montevideo.

Response 3: Thank you for your suggestion, but S. Typhimurium is the abbreviation of Salmonella enterica serovar Typihmurium. Therefore, we represented full name of these species to avoid confusion (L 94-95).

Comments 4: 3.- Novel gene markers are important to describe for these bacterial species described. Because for Listeria and Lactobacillus for example only a short information is mentioned.

Response 4: Thank you for your insightful comment. The importance of the development of gene markers for Listeria and Lactobacillus was indicated in the text (L 194-197, L 221-224).
